# Writing Histone Monoubiquitination in Human Malignancy—The Role of RING Finger E3 Ubiquitin Ligases

**DOI:** 10.3390/genes10010067

**Published:** 2019-01-18

**Authors:** Deborah J. Marsh, Kristie-Ann Dickson

**Affiliations:** Translational Oncology Group, School of Life Sciences, Faculty of Science, University of Technology Sydney, Ultimo, NSW 2007, Australia; Kristie-Ann.Dickson@uts.edu.au

**Keywords:** E3 ligase, RING finger, monoubiquitination, histone, chromatin, RNF20, RNF40, polycomb repressor complex 1, RING1B, BRCA1

## Abstract

There is growing evidence highlighting the importance of monoubiquitination as part of the histone code. Monoubiquitination, the covalent attachment of a single ubiquitin molecule at specific lysines of histone tails, has been associated with transcriptional elongation and the DNA damage response. Sites function as scaffolds or docking platforms for proteins involved in transcription or DNA repair; however, not all sites are equal, with some sites resulting in actively transcribed chromatin and others associated with gene silencing. All events are written by E3 ubiquitin ligases, predominantly of the RING (really interesting new gene) finger type. One of the most well-studied events is monoubiquitination of histone H2B at lysine 120 (H2Bub1), written predominantly by the RING finger complex RNF20-RNF40 and generally associated with active transcription. Monoubiquitination of histone H2A at lysine 119 (H2AK119ub1) is also well-studied, its E3 ubiquitin ligase constituting part of the Polycomb Repressor Complex 1 (PRC1), RING1B-BMI1, associated with transcriptional silencing. Both modifications are activated as part of the DNA damage response. Histone monoubiquitination is a key epigenomic event shaping the chromatin landscape of malignancy and influencing how cells respond to DNA damage. This review discusses a number of these sites and the E3 RING finger ubiquitin ligases that write them.

## 1. Introduction

Histones are basic proteins, with histone H2A, H2B, H3 and H4 constituting the core structure of the nucleosome as an octamer complex around which 147 bp of DNA wrap. An H1 linker protein is associated via DNA entry and exit sites [1,2,3]. N-terminal tails of histones in the nucleosome protrude from this structure and undergo a range of covalent modifications including acetylation, methylation, SUMOylation, phosphorylation, protein isomerisation and ubiquitination [3,4,5]. These structures comprise chromatin and are important for condensing or packaging DNA into higher order chromatin structures. Histone post-translational modifications (PTMs) act as docking sites or scaffolds for proteins that work together to direct cellular transcriptional programmes and the response to DNA damage [6]. Emphasising the importance of these fundamental roles, histones are encoded by multiple genes, including as parts of gene clusters [7,8]. 

The PTM of ubiquitination can take the form of mono-, i.e., covalent attachment of a single ubiquitin to specific lysines for purposes such as gene regulation, or polyubiquitination, where chains of end-to-end ubiquitin molecules form at lysines to mark the protein for degradation via the 26S proteasome requiring ATP hydrolysis [9]. Monoubiquitination is a major cellular event, in yeast shown to account for over half of all conjugated ubiquitin [10]. The most abundantly monoubiquitinated proteins within the nucleus of mammalian cells are histones [11,12]. At 8.5 kDa (76 residues), monoubiquitination of specific lysines on histone tails represents one of the larger histone PTMs relative to modifications such as methylation and acetylation [3,13]. As for polyubiquitination, the ubiquitin enzyme cascade required for monoubiquitination requires three key enzymes [14,15]. In an ATP-dependent reaction, the ubiquitin activating enzyme (E1) forms a thiol-ester bond between its active cysteine site and the carboxy-terminal glycine of ubiquitin. Ubiquitin is then transferred to a cysteine residue of a ubiquitin-conjugating enzyme (E2). Ubiquitin ligases (E3) are critical for substrate recognition, recruiting the E2 ubiquitin conjugate and catalysing substrate ubiquitination [15,16,17,18]. E2 enzymes can interact with multiple E3 enzymes, and, conversely, E3s can interact with different E2s, leading to different ubiquitin linkages and functional outcomes [19,20]. 

Humans have very few E1s, around 40 E2s and as many as 600–1000 E3 enzymes important for ensuring substrate recognition [14,15,21,22]. E3 ligases are classified as either HECT (homologous to E6-AP carboxy terminus) domain or RING (really interesting new gene) domain ligases, the latter being by far the more prevalent [18]. The RING domain is zinc finger domain-like, with 40–60 residues of a motif resembling Cys-X_2_-Cys-X_9-39_-Cys-X_1-3_-His-X_2-3_-Cys-X_2_-Cys-X_4-48_-Cys-X_2_-Cys (X represents any residue and His and Cys can be exchanged) [18,22,23]. It is this domain that provides a docking surface for the E2-ubiquitin conjugate [22].

RING finger ubiquitin protein ligases have roles in both the maintenance of the healthy cell and the oncogenic transformation, including genomic integrity and DNA damage and repair responses [15]. Numerous well-known proteins associated with malignancy function as RING finger E3 ligases, including tumour suppressors BRCA1 [24,25] and the FANC core complex [26], as well as the oncogene MDM2 [27,28]. RING finger E3 ligases also have a specific role in writing histone monoubiquitination, in this way having major influence on the chromatin landscape.

Sites of monoubiquitination of specific lysines on core histone tails will be discussed in this review. These include lysines 34 (K34) [29], 120 (K120, H2Bub1) and 125 (K125) on histone H2B [30,31,32,33,34]; lysines 13 (K13), 15 (K15), 119 (K119, H2AK119ub1), 127 (K127) and 129 (K129) on histone H2A [11,35,36,37,38] and lysine 31 (K31) on histone H4 [39,40] (Figure 1). Given the high levels of monoubiquitination of histone H2A and H2B in the cell, respectively 5–15% and 1-1.5/2%, the discussion of H2AK119ub1 and H2Bub1 will form a larger part of this review [12,41,42]. Core histone H3, the linker histone H1 and H2AX have also been reported to be modified by ubiquitin (reviewed in [42]). While little is known about the effect of some of these ubiquitination events, others have been shown to have opposing effects in the context of their association with either active or silent chromatin, impacting on gene transcription [31,43,44]. This fascinating dichotomy possibly relates to positioning within the nucleosome structure. These ubiquitinated histone sites and the RING finger E3 ligases that write them, including Ring Finger Protein 20 (RNF20), RNF40, RING 1A, RING 1B, MSL2, RNF8, RNF168 and Cullin 4A (CUL4A), are the focus of this review (Table 1).

## 2. Histone H2B Monoubiquitination and the E3 RING Finger Ligase Complex RNF20-RNF40

Histone H2B monoubiquitination was first discovered in yeast, occurring on lysine 123 and carried out by the E3 RING finger ubiquitin ligase Bre1 [45,46]. In 1980, histone H2B was shown to be ubiquitinated in mouse and man [12]. In both yeast and humans, RAD6A is the E2-conjugating enzyme, also referred to as the ubiquitin-conjugating enzyme 2A (UBE2A) [47]. A second E2, UBE2E1 (also known as UBCH6), for H2Bub1 has been reported [48,49]. In humans, H2Bub1 occurs at lysine 120 of histone H2B and functions to alter chromatin compaction, leading to open fibres that are more readily accessible to transcription factors and DNA repair proteins [50]. Lysine 120 of histone H2B resides at the interface of adjacent nucleosomes, with the addition of the bulky ubiquitin molecule likely interfering with nucleosome stacking and therefore chromatin configuration. This effect, however, is more than purely positional as the addition of the even bulkier modification of SUMO to this site does not behave in the same manner [50,51].

A number of different E3 ubiquitin ligases have been implicated as being able to monoubiquitinate histone H2B. These include MDM2 [34], the BRCA1-BARD1 complex [52,53] and BAF250B [54]. It is generally accepted, however, that the RNF20-RNF40, an orthologue of yeast Bre1, is the main RING finger E3 ligase complex capable of writing H2Bub1 in humans [30,32,33,47,48,55]. RNF20 and RNF40 form a heterotetrameric complex with two copies of each polypeptide [48]. While the key functional unit of this complex has been reported as RNF20 [56], it would appear that both subunits are likely important to the stability and function of this complex given that downregulation of either RNF20 or RNF40 leads to depletion of the other as well as to depletion of H2Bub1 [55,57,58]. Furthermore, overexpression of RNF20 has been shown to increase levels of H2Bub1 and subsequently histone methylation at lysines 4 and 79 on histone H3 as well as to stimulate the expression of HOX genes, a group of homeotic genes that are master controllers of embryonic development [48,56]. 

The importance of histone crosstalk involving this modification was reported in yeast, whereby monoubiquitination of yeast histone H2B K123 was shown to be required for methylation of histone H3K4 [59,60]. In humans, H2Bub1 is a central modification, able to recruit methyltransferase complexes such as COMPASS and DOTL1, leading to H3K4 di- and tri-methylation, as well as H3K79 tri-methylation (reviewed in [30]). This influence on histone methyltransferase events is not unique to H2Bub1, as discussed later in this review. It is important to note the dynamic nature of monoubiquitination, and while they will not be discussed further in this review, numerous deubiquitinases (DUBs), including USP7 and USP44, have been reported to erase H2Bub1 (reviewed in [30,32,33]).

### 2.1. RNF20-RNF40 and H2Bub1 in Transcriptional Elongation and DNA Damage

The interaction between RNF20-RNF40 and the PAF1 (RNA polymerase II-associated factor 1) complex is thought to be responsible for the increase in H2Bub1 at the coding regions of transcriptionally active genes [47]. The PAF1 complex is comprised of multiple subunits, a number of which, including the tumour suppressor CDC73, have been shown to interact with RNF20-RNF40 [30,57,61]. When the cell receives a signal to activate gene expression, cyclin-dependent kinase 9 (CDK9) phosphorylates both the H2Bub1 E2-conjugating enzyme UBE2A and Ser2 of the carboxy-terminal domain of RNA polymerase II. This creates a binding pocket for WAC (WW domain containing adaptor with coiled-coil) that associates with RNF20-RNF40 to enable H2Bub1 [62]. The RNF20-RNF40 association with the PAF1 complex that couples with RNA polymerase II at chromatin promotes transcriptional elongation (reviewed in [30]). The chromatin remodelling factor FACT is recruited, enabling removal of the H2B-H2A core histone dimer from the nucleosome structure, eliminating the physical block to RNA polymerase II and allowing transcriptional elongation to proceed (reviewed in [30]) [49,63]. Perhaps curiously, however, down-regulation of RNF20 or RNF40 does not affect the transcription of the majority of genes [64,65]. This introduces the concept that these E3 ligases may function as ‘selective’ tumour suppressors. It is possible that the location of genes throughout the genome plays a part in this apparent anomaly. One key gene that RNF20 depletion does reduce is the tumour suppressor p53 [64].

In response to DNA damage, the ATM (ataxia telangiectasia mutated) kinase phosphorylates RNF20 and RNF40 that are then directed to sites of double strand breaks (DSBs) where they function to monoubiquitinate histone H2B at lysine 120, facilitating open chromatin accessible by DNA repair proteins [55]. As part of this process, repair proteins, including those involved in homologous recombination (RAD51, BRCA1 and BRCA2) as well as those involved in non-homologous recombination (XRCC4 and Ku80), are recruited [55,66,67]. RNF20 is key in these processes, recruiting the chromatin remodelling factor SNF2h and establishing the importance of the RNF20 RING finger E3 ligase in shaping the chromatin landscape [67]. Furthermore, RNF20 has been shown to interact with p53, the guardian of the genome, associating with p53 at the promoters of p53 target genes [56]. Interestingly, a recent study has reported that the known apoptotic mark serine 14 on histone H2B is phosphorylated (H2BS14p) in response to DSBs induced by ionising radiation in an ATM-dependent manner, marking transcriptionally silent nucleolar chromatin [68]. 

### 2.2. RNF20, RNF40 and H2Bub1 in Cancer

Immunohistochemical assessment of overall, or global, levels of H2Bub1 has shown loss of H2Bub1 in a range of primary tumours, including breast, colon, lung, ovarian, parathyroid, gastric and testicular cancer [58,65,69,70,71,72,73,74,75]. Retention of H2Bub1 in non-cancer cells has been shown for a number of tissue types, including normal colonic mucosa and normal breast tissue adjacent to tumours [65,73]. In some tumour types, such as breast, loss of global levels of H2Bub1 has been associated with advancing tumour progression; however, this is not true for all malignancies, with H2Bub1 loss observed in both early and late stages of high-grade serous ovarian cancer [58,65]. Global loss of H2Bub1 has also been correlated with poor patient survival in colorectal cancer [73]. Loss of H2Bub1 has been associated with mutation of the tumour suppressor CDC73, a member of the PAF1 complex, in parathyroid tumours [57]. The clear mechanism underpinning this loss across multiple tumour types remains to be elucidated.

A small number of studies have assessed RNF20 in primary tumours. In a study of 424 high-grade serous ovarian cancers, loss of RNF20 was only seen in ~6% of tumours and, perhaps surprisingly, did not correlate with loss of H2Bub1 [58]. In apparent contrast, *RNF20* and *RNF40* mRNA levels were reduced in colons from patients with ulcerative colitis [71]. Further in this study, *RNF20* and *RNF40* levels were inversely correlated with mRNA levels of *IL6* and *IL8* inflammatory cytokines, causing the authors to speculate that constitutively reduced levels of *RNF20* and *RNF40* leading to reduced levels of H2Bub1 may increase the risk of developing certain chronic inflammatory diseases, at least those associated with the colon [71]. It would be interesting to see whether decreases at the gene expression level are recapitulated at the protein level for RNF20 and RNF40 in colorectal tissue associated with inflammatory conditions. Lastly, in a study of clear cell renal cell carcinoma, loss of RNF20 detected in primary tumours by immunohistochemistry was a marker of poor prognosis [76].

### 2.3. RNF20, RNF40 and H2Bub1 Display Both Tumour-Suppressive and Oncogenic Functions

The *RNF20* promoter has been reported to be hypermethylated in primary breast cancer cells [64,77] and mutated at low frequency in colorectal cancer [78,79]. *RNF20* mRNA levels are also reduced in metastatic prostate cancer cells compared to benign disease and are lower in testicular germ cell cancer seminoma compared to normal testis [75,80]. Genomic loss of RNF20 has been reported in pre-invasive dysplastic airway lesions [81]. It will be interesting to observe whether these genetic and epigenetic changes impact upon the protein levels and function of RNF20 and RNF40. Furthermore, overexpression of RNF20 in renal cell cancer cell lines led to a decrease in proliferation, while suppression of RNF20 led to increased proliferation [76]. Another way that RNF20 may function as a tumour suppressor is by stopping recruitment of the transcription elongation factor TFIIS to chromatin. TFIIS works by releasing stalled RNA polymerase II, and its inhibition works to impede the expression of oncogenes such as *MYC* and *FOS* that normally reside in regions of compacted chromatin [77].

While most of the literature presented thus far would suggest that H2Bub1 and its E3 ligases function as tumour suppressors, there are a number of studies that suggest that higher levels and/or activity of these factors may actually promote tumorigenesis, i.e., have an oncogenic function (reviewed in [82]). Down-regulation of RNF20 has been shown to lead to the migration of MCF10A breast epithelial cells, as well as anchorage-independent growth of NIH3T3 cells [64]. In an opposite fashion, upregulation of RNF20 leading to increased Hox gene expression may also contribute to a malignant phenotype, and knockdown of RNF20 in the breast cancer cell line MCF7 led to reduced proliferation [48,83]. Furthermore, a recent study of different subtypes of breast cancer has shown that whether RNF20 and H2Bub1 inhibit or enhance cellular proliferation and migration is entirely dependent on the subtype. Here, silencing of RNF20 led to increased proliferation and migration in basal-like breast tumours, likely via upregulation of inflammatory cytokines, while silencing of RNF20 in luminal breast cancer cells decreased proliferation and migration, compromising transcription of the estrogen receptor [84]. In another apparent contrast, loss of RNF20 has been linked with an inflammatory phenotype in colorectal cancer [71], while loss of RNF40 would appear to have a protective effect against the development of an inflammatory phenotype in the same cancer [85], both involving NF-κB signalling. Fundamentally, whether RNF20, RNF40 and H2Bub1 predominantly drive or inhibit proliferative or inhibitory phenotypes via remodelling of the chromatin landscape might be influenced by models chosen for study and/or be cell-type- and disease-specific, likely underpinned by specific transcriptional activity. Furthermore, the effect of modulation of these proteins in the context of DNA damage and repair is currently unclear.

### 2.4. Non-Histone Substrates of RNF20 and RNF40

While there is a large and growing body of literature investigating histone H2B lysine 120 as a substrate of the RNF20-RNF40 complex, other substrates of these E3 ligases have also been reported. At present, it cannot be excluded that some tumour-suppressive or oncogenic functions of RNF20 and RNF40 might occur via non-histone substrates. RNF20-RNF40 is known to monoubiquitinate the motor protein Eg5 that has roles in spindle assembly during mitosis [86]. RNF20 has also been reported to polyubiquitinate the ErbB3 receptor binding protein Ebp1 [83]. Staring, the rat orthologue of RNF40, has been shown to polyubiquitinate Syntaxin 1, which is part of the neurotransmitter release machinery with links to learning and memory behaviour, facilitating its degradation via the ubiquitin proteasome [87]. It is currently unclear whether RNF20 and RNF40 may partner with different E2 enzymes for the purpose of ubiquitinating these non-histone substrates.

## 3. Histone H2A Monoubiquitination

The first protein identified over 40 years ago to be ubiquitinated was histone H2A [88]. Along with histone H2B, histone H2A is one of the most abundantly ubiquitinated nuclear proteins, with 5–15% of histone H2A being monoubiquitinated in human cells [11]. There is more than one site that undergoes monoubiquitination on histone H2A. Unlike H2Bub1, H2AK119ub1 does not impact on fibre compaction, likely related to its nucleosomal position [89]. Contrasting with H2Bub1, H2AK119ub1 is associated with gene silencing [43,44]. The mechanism underpinning the repression of gene expression is the prevention of FACT recruitment that otherwise relieves nucleosome barriers to RNA polymerase II-related transcriptional elongation [49,90]. Furthermore, H2AK119ub1 inhibits the transcriptionally active methylation marks H3K4me2 and me3 [91]. High levels of expression of H2AK119ub1 have been associated with a worse prognosis in some tumours [92].

### 3.1. H2AK119ub1 is Written by the Polycomb Repressive Complex 1 

H2AK119ub1 is predominantly written by RING finger E3 ligases as part of the polycomb group (PcG) proteins, specifically polycomb repressive complex 1 (PRC1) [35,36,93,94,95]. Polycomb repressive complex 2 (PRC2) is comprised of four main components—EZH1/2, SUZ12, EED and RbAp46/48 (also referred to as RBBP7/4). It is through the methyltransferase activity of EZH1 and EZH2 (enhancer of zeste 1/2 polycomb repressive complex 2 subunit) that histone H3 at K27 is di- and tri-methylated (H3K27me2/3) [36]. PcG proteins work with their antagonists in the trithorax family during embryonic development and in adult cells to maintain the expression of key developmental genes important for the regulation of cell fate [96]. 

The RING domain ligases constituting PRC1 include RING1B (also known as Ring2/Rnf2), RING1A and BMI1 [44,94,95]. RING1B appears to be the dominant RING finger E3 ligase [44]. Through this mechanism of monoubiquitination, the PRC1 is involved in X chromosome inactivation [97], Hox gene silencing [95] and polycomb target gene silencing [94]. Loss of RING1A, RING1B or BMI1 leads to a global reduction of H2AK119ub1, cementing the importance of these PRC1 subunits as major E3 ligases for this histone modification [42]. 

RING1B associates with other repressive complexes, including E2F-6.com-1, which is involved in the silencing of E2F and MYC responsive genes in quiescent cells and the FBXL10-BcoR complex that likely also monoubiquitinates histone H2A at lysine 119 (reviewed in [44]). Monoubiquitination of the histone variant H2A.Z at K120 or K121 also involves RING1B as the E3 ligase [98]. Like many E3 ligases, RING1B has more than one substrate and has been shown to ubiquitinate the tumour suppressor p53, marking it for degradation [99]. The Cullin4B-E3 ligase complex, CRLB4, is physically associated with PRC2 and has been shown to catalyse H2AK119ub1 [100,101]. The RING finger domain E3 ligase TRIM37 has also been shown to catalyse H2AK119ub1, again interacting with the PRC2 complex [102]. Interestingly, the E3 ligase MDM2 interacts with the PRC2 complex, enhancing H2AK119ub1 [103]. Also able to monoubiquitinate histone H2A is the HECT domain E3 ligase LASU1 [104].

### 3.2. RING1B in Malignancy

RING1B is highly expressed in a number of human malignancies including prostate cancer, pancreatic ductal adenocarcinoma, ovarian cancer and urothelial bladder carcinoma [92,105,106,107]. Overexpression of RING1B has been associated with a poorer prognosis for women with ovarian cancer [105] and shorter survival times for patients with urothelial bladder carcinoma [106].

### 3.3. H2AK119ub1 is Specifically Written by 2A-HUB to a Subset of Chemokine Genes

2A-HUB/hRUL138 is an alternative RING domain E3 ligase recruited by the N-CoR/HDAC1/3 complex to the promoters of some chemokine genes to monoubiquitinate histone H2A at lysine 119 and repress expression of these chemokines [90]. Unlike the case of PRC1, this does not appear to be a global effect; rather, it is specific to this subset of genes that have specific roles in ensuring an immune response to inflammatory stimuli [90]. 

### 3.4. Histones H2A, H2AX and the Response to DNA Damage

As for H2Bub1, histone H2A is monoubiquitinated at sites of DNA damage. RING1B-BMI1 is recruited to sites of DSBs where it catalyses H2A and H2AX monoubiquitination at lysine 119 [42,108,109]. These modifications are thought to be important for the recruitment of the ATM kinase to sites of DSBs, ATM being one of the earliest responders to DNA damage [41,110]. Furthermore, the DNA damage-responsive E3 ubiquitin ligases RNF8 and RNF168 monoubiquitinate histone H2A and H2AX at lysines 13 and 15 [37,111,112,113,114]. Modified sites then act as significant scaffolds for the recruitment of key proteins engaged in the DNA damage response, such as BRCA1 and 53BP1 [42].

As well as being able to function as an E3 ligase RING finger complex for H2Bub1, BRCA1-BARD1 co-operates with the E2 UbcH5c to monoubiquitinate H2A and H2AX [42,52,115]. BRCA1-deficient cells display loss of ubiquitinated histone H2A at pericentromeric tandemly repeated satellite DNA [116]. The BRCA1-BARD1 complex ubiquitinates histone H2A at lysines 127 and 129 [38,114]. 

## 4. Other Known Lysine Sites of Histone Tail Monoubiquitination and Their E3 RING Finger Ligases

Other sites of histone monoubiquitination have been reported in human cells but remain to be extensively investigated. K125 on the histone H2B tail is monoubiquitinated when K120 is mutated in vitro; however, there is no evidence to date that H2BK125ub1 occurs in vivo [34]. Monoubiquitination of histone H2B on lysine 34 (H2BK34ub1) is written by the RING finger E3 ligase MSL2 that is part of the MOF-MSL complex [29]. This complex consists of four subunits (MOF-MSL1-MSL2-MSL3) [117], with MSL1-MSL2 being required for optimal E3 ubiquitin ligase activity. Like H2Bub1, H2BK34ub1 facilitates transcriptional elongation by RNA polymerase II [118].

Interestingly, MOF (also known as KAT8 or MYST Histone Acetyltransferase 1) functions as part of the MOF-MSL complex to acetylate histone H4 at lysine 16, giving this complex a dual role of ubiquitination and acetylation [119,120]. Providing early evidence of a complex network of histone E3 RING finger ligases, investigators have shown that MSL2 is involved in the recruitment of the H2Bub1 E3 ubiquitin ligases RNF20 and RNF40 to chromatin [29]. Fitting the paradigm of E3 RING finger ligases having multiple substrates, MSL2 is known to ubiquitinate the tumour suppressor p53, but does so at different sites to the better-known p53 E3 ligase MDM2 [121,122]. As is H2Bub1, H2BK34ub1 is involved in histone cross-talk, regulating H3K4me3 and H3K79me2 [29].

Monoubiquitination of histone H4 at lysine 31 (H4K31ub1), like H2Bub1, is associated with a more open chromatin configuration conducive to the promotion of transcription [39,40]. The cullin 4A (CUL4A) RING finger E3 ubiquitin ligase has been associated with these ubiquitination events, albeit at much lower levels than H2Bub1 and H2AK119ub1 [40,123]. Furthermore, CUL4A-mediated ubiquitination of H4K31 is also associated with accumulation of the active methylation marks H3K4me3 and H3K79me2 [40].

## 5. Therapeutic Targeting of E3 Ligases that Write Histone Monoubiquitination

Given the significant role of E3 ubiquitin ligases in substrate recognition, their value as therapeutic targets for diseases such as cancer are being considered [124,125]. Although no drugs have yet been developed with the goal of impacting specifically upon E3s that write histones, knowledge gleaned from developments to target non-histone E3 writers should be considered. While the E3 RING finger ubiquitin ligase MDM2 is reported to write H2Bub1 in vivo [34], it is primarily known as the enzyme that polyubiquitinates the guardian of the genome tumour suppressor p53, marking it for degradation via the 26S proteasome [126,127]. Small molecule inhibitors that disrupt the interaction of MDM2 and p53, including the *cis*-imidazoline compounds nutlin and the nutlin derivative RG7388 (idasanutlin, RG), are being developed for the treatment of a number of cancers [128,129,130]. In preventing polyubiquitination of p53 and thus its degradation, wild-type p53 levels are increased, in turn promoting apoptosis.

Other RING finger E3 ligases being considered as therapeutic targets for a range of diseases include cereblon, XIAP, IAP, VHL complex and Parkin [125]. An important step that necessarily underpins drug design targeting all of the E3 ligases that monoubiquitinate histones will be a sound understanding of their non-histone substrates [16,125]. In this vein, it is interesting to speculate that disruption of E3 ubiquitin ligases that function as complexes, for example the RNF20-RNF40 interaction, may also have merit when considering strategies for therapeutic targeting.

A deeper understanding of the fact that single sites of histone monoubiquitination seem to be written by multiple E3 ubiquitin ligases will be important in order to maximise therapeutic targeting opportunities. Currently, the reasons for this apparent redundancy are poorly understood but would seem to highlight the importance of these ubiquitination events to decisions regarding cellular maintenance and fate, including those in response to DNA damage. It is also likely that this function of some of these E3 ligases may be specific to cell type, disease state and/or developmental stage.

## 6. Conclusions

The role of multiple E3 RING finger ubiquitin ligases in writing monoubiquitination at specific lysine residues of histone tails is clearly fundamental to RNA polymerase II-based transcriptional elongation and the DNA damage response. Cancer is frequently described as a disease of aberrant transcription, highlighting the potential for the development of new therapeutic strategies by targeting these key enzymes functioning as part of the transcriptional machinery. There is a high likelihood that additional sites of histone monoubiquitination and their writer enzymes will be identified. Whether these will affect global levels of histone monoubiquitination or offer new levels of complexity by being directed towards modifying the expression of only certain functional classes of genes remains to be determined.

## Figures and Tables

**Figure 1 genes-10-00067-f001:**
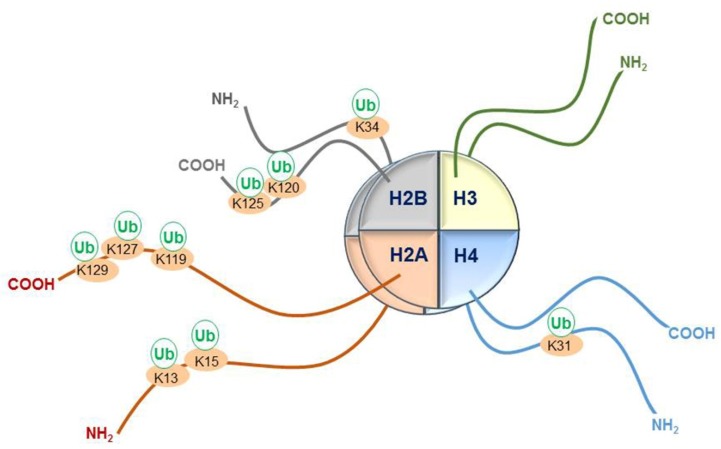
Schematic representation of sites of monoubiquitination at lysines (K) on histone tails protruding from the core octamer structure of nucleosomes. Monoubiquitination at K34 and K120 on histone H2B, as well as at K31 on histone H4, has been associated with transcriptional elongation. Monoubiquitination of K119 on histone H2A is written by an E3 ligase in polycomb repressor complex 1 and is associated with inactive chromatin. K13 and K15 on the histone H2A tail are written by E3 ligases associated with DNA damage. K127 and K129 on histone tail H2A are ubiquitinated by E3 ligase activity of the Breast Cancer 1 (BRCA1)- BRCA1-Associated RING Domain Protein 1 (BARD1) complex. It is unclear whether K125 is monoubiquitinated in vivo. Monoubiquitination at K31 on histone H4, as well as at K34 and K120 on histone H2A, participates in histone crosstalk to promote the methylation of K4 and K79 on histone H3. Ub: Ubiquitin.

**Table 1 genes-10-00067-t001:** Sites of histone ubiquitination and the E3 ligases responsible for writing them.

Histone Modification	E3 Ligases Participating in Histone Monoubiquitination
Histone H2B:	
H2B K120ub1 (H2Bub1)	RNF20-RNF40, MDM2, BRCA1-BARD1, BAF250B
H2B K34ub1	MSL2
Histone H2A:	
H2A K119ub1	Polycomb repressor complex 1 (RING1A-RING1B-BMI1),
	CRLB4, TRIM37, LASU1, MDM2, 2A-HUB/hRUL138
H2A K13ub1, K15ub1	RNF8, RNF168
H2A K127ub, K129ub	BRCA1-BARD1
Histone H2A.Z:	
H2A.Z K120ub, K121ub	RING1B
Histone H2AX:	
H2AX K119ub1	RING1B-BMI1
H2AX K13ub1, K15ub1	RNF8, RNF168
H2AX K127ub, K129ub	BRCA1-BARD1
Histone H4:	
H4 K31ub1	CUL4A

RNF20-RNF40: Ring Finger Protein 20 - 40; MDM2: Mouse Double Minute 2; BRCA1-BARD1: Breast Cancer 1 – BRCA1- Associated RING Domain Protein 1; BAF250B: also known as ARID1B, AT-Rich Interaction Domain 1B; MSL2: Male-Specific Lethal 2 Homolog; RING1A-RING1B-BMI1: Ring Finger 1A-Ring Finger 1B-B Lymphoma Mo-MLV Insertion Region 1 Homolog; CRLB4: Cullin4B-E3 ligase complex; TRIM37: Tripartite Motif Containing 37; LASU1: also known as E3^Histone^; 2A-HUB/hRUL138: H2A monoubiquitination ligase/human RNA-binding ubiquitin ligase of 138 kDa; CUL4A: Cullin 4A.

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
