# Peer review of "Writing Histone Monoubiquitination in Human Malignancy—The Role of RING Finger E3 Ubiquitin Ligases"

_genes, 2019, doi:10.3390/genes10010067_

Round 1

Reviewer 1 Report

This review addresses the role of ubiquitin targeting in controlling transcription through histone ubiquitination, with a special focus on oncogenic/anti-oncogenic functions. Ubiquitin modification of proteins controls almost all the cellular processes and better understanding the underlying mechanisms is essential for the development of therapeutic strategies, hence the importance of this topic. 

This review is well written and easy to follow, and the presence of a Table summarizing what is known about the sites of ubiquitination and the E3 ligases involved is useful. I do have some suggestions to ameliorate some paragraphs and/or to precise some points.

1.     In the introduction section, I suggest to clearly precise that ubiquitination may be monoUb or polyUb in chains, the former being more a regulatory signal not affecting the stability of the modified protein and the latter being generally associated with the hydrolysis of the protein targeted by a polyUb chain. It is also important to precise that monoubiquitination is a major Ub event (it represent roughly 50% of the ubiquitinated proteins in yeast (Ziv et al Mol Cell Proteomics 2011). It is also important to explain that an E3 can work with several E2s (and vice versa), which ends up with a panel of different signals and fates for the target. 

2.     Histones ubiquitination can use several E3 ligases for the same spot, at least for H2B K120ub1 and H2A K119ub1. This raises the following question: why are so many E3s needed? Several hypotheses can be drawn (redundancy, cell type specificity, …) but this is a key point that future investigations will have to solve before therapeutic strategies could be proposed. This aspect should be included in the review, either in individual paragraph or as a new paragraph “future directions”. 

3.     Paragraph 2.4: It is important to precise that the different signals (mono vs. polyUb) the RNF20-RNF40 E3 ligases build on substrates is probably due to the presence of different E2 enzymes as the E2s determine the type of ubiquitination for RING finger E3 ligases.

4.     Line 79: One should read Rad6A. There are 2 close isoforms, RAD6A/UBE2A and RAD6B/UBE2B and each of them may have differential specificity. Recently, RAD6B/UBE2B has been shown to monoubiquitinate histones H2A and H2B (Guo, Cell cycle 2018) in cooperation with the E3 ligase RNF8. Similarly, H2B is modified by RAD6B and RNF168 (Liu et al J Cell Science 2013). These data have to be included in the text and in the Table 1. 

Author Response

Reviewer One:

We would like to thank this reviewer for their helpful comments and address them below in order. All changes to the submitted manuscript are highlighted in yellow.

In the introduction section, I suggest to clearly precise that ubiquitination may be monoUb or polyUb in chains, the former being more a regulatory signal not affecting the stability of the modified protein and the latter being generally associated with the hydrolysis of the protein targeted by a polyUb chain. It is also important to precise that monoubiquitination is a major Ub event (it represent roughly 50% of the ubiquitinated proteins in yeast (Ziv et al Mol Cell Proteomics 2011). It is also important to explain that an E3 can work with several E2s (and vice versa), which ends up with a panel of different signals and fates for the target. 

The following sentence has been added to the Introduction to more clearly define mono- versus poly-ubiquitination. It required minor modification of a nearby sentence as follows:

The PTM of ubiquitination can take the form of mono- ,i.e. covalent attachment of a single ubiquitin to specific lysines for purposes such as gene regulation, or polyubiquitination where chains of end-to-end ubiquitin molecules form at  lysines to mark the protein for degradation via the 26S proteasome requiring ATP hydrolysis [9]. …  The most abundantly monoubiquitinated proteins within the nucleus of mammalian cells are histones [10, 11]. At 8.5 kDa (76 residues), monoubiquitination of specific lysines on histone tails represents one of the larger histone PTMs relative to modifications such as methylation and acetylation [3, 12].”

The following sentence and reference have been added to the Introduction to point out that monoubiquitination is a major cellular event:

“Monoubiquitination is a major cellular event, in yeast shown to account for over half of all conjugated ubiquitin [10]

The following sentence and references have been added to the Introduction to explain that E3s and E2s are promiscuous with each other.

“E2 enzymes can interact with multiple E3 enzymes, and conversely, E3s can interact with different E2s, leading to different ubiquitin linkages and functional outcomes [19, 20].”

 2. Histones ubiquitination can use several E3 ligases for the same spot, at least for H2B K120ub1 and H2A K119ub1. This raises the following question: why are so many E3s needed? Several hypotheses can be drawn (redundancy, cell type specificity, …) but this is a key point that future investigations will have to solve before therapeutic strategies could be proposed. This aspect should be included in the review, either in individual paragraph or as a new paragraph “future directions”. 

An additional paragraph has been added to Section 5 “Therapeutic targeting of E3 ligases that write histone monoubiquitination” to specifically address this point as follows:

A deeper understanding of the fact that single sites of histone monoubiquitination seem to be written by multiple E3 ubiquitin ligases will be important in order to maximise therapeutic targeting opportunities. Currently, the reasons for this apparent redundancy are poorly understood, but would seem to highlight the importance of these ubiquitination events to decisions regarding cellular maintenance and fate, including in response to DNA damage. It is likely also that this function of some of these E3 ligases may be specific to cell type, disease state and/or developmental stage.”

3. Paragraph 2.4: It is important to precise that the different signals (mono vs. polyUb) the RNF20-RNF40 E3 ligases build on substrates is probably due to the presence of different E2 enzymes as the E2s determine the type of ubiquitination for RING finger E3 ligases.

This is a very good point. An additional sentence has been added to this section to specifically address this possibility as follows:

It is currently unclear whether RNF20 and RNF40 may partner with different E2 enzymes for the purpose of ubiquitinating these non-histone substrates”.

 4. Line 79: One should read Rad6A. There are 2 close isoforms, RAD6A/UBE2A and RAD6B/UBE2B and each of them may have differential specificity. Recently, RAD6B/UBE2B has been shown to monoubiquitinate histones H2A and H2B (Guo, Cell cycle 2018) in cooperation with the E3 ligase RNF8. Similarly, H2B is modified by RAD6B and RNF168 (Liu et al J Cell Science 2013). These data have to be included in the text and in the Table 1. 

 - “RAD6” has been corrected to “RAD6A”

- We have not added the Guo et al. reference into the text or to the table, largely because the data presented is in mice only, they have not shown the actual sites of monoubiquitination in histones H2A and H2B (so we are unable to discuss this in the context of H2Bub1 or H2AK119ub1 for example) and their study pertains to spermatogenesis only. Relation of the data to humans and the specific lysine sites would therefore be speculative and we would prefer not to include this at this stage.

- We would prefer also to not include the Liu et al. reference as the sites of ubiquitination are just not presented and so not a fit with the specific nature of the ubiquitination sites that we have recorded in the table, or how we have presented our review as a whole. We hope this is amenable to this assessor.

Reviewer 2 Report

The review presents a fantastic overview of the ubiquitination of histones and was a pleasure to read. My only comment is to include additional comments to section 3.4 in light of the recent article by Pefani et al. EMBO 2018 that discussed the impact of phosphorylation of H2B in the DDR. The site was pS14-H2B and neighbours K15, an important mono-ub site that is discussed elsewhere. 

Author Response

Reviewer Two:

Comments and Suggestions for Authors

The review presents a fantastic overview of the ubiquitination of histones and was a pleasure to read. My only comment is to include additional comments to section 3.4 in light of the recent article by Pefani et al. EMBO 2018 that discussed the impact of phosphorylation of H2B in the DDR. The site was pS14-H2B and neighbours K15, an important mono-ub site that is discussed elsewhere. 

We would like to thank this reviewer for their positive comments. We have included the reference as suggested, adding text to Section 2.1 where histone H2B modifications are discussed in response to DNA damage (section 3.4 pertains to histone H2A).

“Interestingly, a recent study has reported that the known apoptotic mark serine 14 on histone H2B is phosphorylated (H2BS14p) in response to DSBs induced by ionizing radiation in an ATM-dependent manner, marking transcriptionally silent nucleolar chromatin”.